

# Compatibility between object size and response side in grasping: the left hand prefers smaller objects, the right hand prefers larger objects

Christian Seegelke[1,2] and Peter Wühr[3]

[1] Biopsychology and Cognitive Neuroscience, Faculty of Psychology and Sport Sciences, Bielefeld University, Bielefeld, Germany
[2] Center of Excellence Cognitive Interaction Technology (CITEC), Bielefeld University, Bielefeld, Germany
[3] Institute of Psychology, TU Dortmund University, Dortmund, Germany

## ABSTRACT

It has been proposed that the brain processes quantities such as space, size, number, and other magnitudes using a common neural metric, and that this common representation system reflects a direct link to motor control, because the integration of spatial, temporal, and other quantity-related information is fundamental for sensorimotor transformation processes. In the present study, we examined compatibility effects between physical stimulus size and spatial (response) location during a sensorimotor task. Participants reached and grasped for a small or large object with either their non-dominant left or their dominant right hand. Our results revealed that participants initiated left hand movements faster when grasping the small cube compared to the large cube, whereas they initiated right hand movements faster when grasping the large cube compared to the small cube. Moreover, the compatibility effect influenced the timing of grip aperture kinematics. These findings indicate that the interaction between object size and response hand affects the planning of grasping movements and supports the notion of a strong link between the cognitive representation of (object) size, spatial (response) parameters, and sensorimotor control.

Corresponding author
Christian Seegelke,
christian.seegelke@uni-bielefeld.de

## INTRODUCTION

According to ''A Theory of Magnitude'' (ATOM), there exists a generalized magnitude system in the brain, that processes quantities such as space, size, number, time, and other magnitudes using a common neural metric (*Walsh, 2015*; *Walsh, 2003*). Several neuroscientific studies have provided evidence for such a shared neural representation by showing activation in overlapping areas within the posterior parietal cortex (PPC) during the processing of different magnitudes (e.g., *Jacob & Nieder, 2009*; *Pinel et al., 2004*; see *Hubbard et al., 2005*; *Bueti & Walsh, 2009* for reviews). On a behavioral level, this notion is supported by studies showing interaction effects between several dimensions addressed in ATOM such as number and space (*Dehaene, Bossini & Giraux, 1993*; *Winter et al., 2015*), number and size (*Henik & Tzelgov, 1982*; *Reike & Schwarz, 2017*), time and space

(*Bonato, Zorzi & Umiltà, 2012*), size and space (*Sellaro et al., 2015*; *Wühr & Seegelke, 2018*), and other magnitudes (*Macnamara, Keage & Loetscher, 2018*). For example, interactions between numbers and space are evident in the "Spatial-Numerical Association of Response Codes (SNARC)" effect. Here, participants typically respond faster with the left hand in response to (relative) small numbers as compared to (relative) large numbers, whereas they respond faster with the right hand in response to (relative) large numbers as compared to (relative) small numbers (e.g., *Dehaene, Bossini & Giraux, 1993*). The SNARC effect has led to the suggestion that numbers are spatially organized along a mental number line (MNL) with small numbers represented to the left and large numbers to the right (*Restle, 1970*; *Dehaene, Bossini & Giraux, 1993*).

According to ATOM, this common representation system evolved through interaction with the environment, as it is through movement that we learn associations between different magnitude domains, for example that larger objects are usually heavier and that it takes more time to cover a larger distance (*Walsh, 2003*; *Binetti et al., 2015*). This assumption is corroborated by the fact that many subregions within the PPC (and particularly in the intraparietal sulcus, IPS) are involved in visuomotor transformations of spatial parameters required for specific motor actions (see *Rizzolatti & Luppino, 2001*; *Gallivan & Culham, 2015* for reviews). Accordingly, this common representation system reflects a functional organization subserving motor control, because the integration of spatial, temporal, and other quantity-related information is fundamental for sensorimotor transformation processes within the PPC (*Crawford, Henriques & Medendorp, 2011*).

The proposal of a strong link between magnitude processing and sensorimotor processes is supported by studies showing direct interactions between magnitude processing and the planning and/ or execution of movements. For example, the SNARC effect is not only expressed in faster reaction times for small-left and large-right associations (e.g., *Dehaene, Bossini & Giraux, 1993*), but also evident in movement execution times (*Fischer, 2003*), and systematic trajectory shifts (*Song & Nakayama, 2008*) in manual reaching tasks.

Besides reaching paradigms, grasping provides a particularly suitable task to study magnitude-motor interactions, since grasping naturally requires the processing of magnitude-related information. Reach-to-grasp movements exhibit a clear spatio-temporal profile that is characterized by a progressive opening of the grip with its peak (i.e., maximum grip aperture) highly correlated with object size, followed by closing of the grip until it matches the object size (*Castiello, 2005*). Grasping tasks have often been employed to study interactions between the processing of numerical magnitudes and size-related motor parameters. In an initial study, *Andres et al. (2004)* showed that participants initiated hand-closing movements faster in response to small numbers and hand-opening movements faster in response to larger numbers. Although the task did not require the grasping of an object, the authors still argued that this interaction arose due to a common magnitude representation for number processing and the computation of an appropriate grip aperture. Subsequent studies provided more conclusive evidence for this claim (*Lindemann et al., 2007*; *Andres et al., 2008*). These studies demonstrated that precision grips are initiated faster in response to small numbers whereas power grips are initiated faster in response to large numbers (*Lindemann et al., 2007*), and that grasping kinematics exhibit increased

grip aperture in the presence of large numbers (*Lindemann et al., 2007*; *Andres et al., 2008*; *Namdar et al., 2014*). Together, these findings strongly suggest that number magnitude processing interacts with the sensorimotor processes involved in shaping the hand grip to object size.

Although most studies have examined interactions between numerical information and motor-related spatial coding (e.g., *Andres et al., 2004*; *Andres et al., 2008*; *Lindemann et al., 2007*; *Namdar et al., 2014*; *Namdar & Ganel, 2018*; *Rugani, Betti & Sartori, 2018*; *Rugani et al., 2017*; *Badets & Pesenti, 2010*; *Ranzini et al., 2011*), similar interactions have also been observed for other magnitude domains. For example, when participants reached out to grasp a wooden block, their movements had a larger grip aperture after reading a word representing a larger object (e.g., apple) than reading a word representing a small object (e.g., grape *Glover et al., 2004*), indicating interactions between (conceptual) object size and reach-to-grasp movements (see also *Gentilucci et al., 2000*; *Glover & Dixon, 2002*).

In the present study, we address two domains whose potential interactions have largely been neglected in the literature, but clearly are relevant for sensorimotor control: physical object size and space. Recently, we demonstrated a SNARC-like compatibility effect between physical object size and (horizontal) response location using a classic stimulus–response (S-R) compatibility task that required a left or right button press in response to a large or small square in each trial (*Wühr & Seegelke, 2018*). Specifically, we found that with the left hand, participants responded faster to small stimuli than to large stimuli, whereas with the right hand, they responded faster to large stimuli than to small stimuli (see *Ren et al., 2011* for similar results). This size-space compatibility effect was also observed when stimulus size was irrelevant for task performance, indicating some degree of automaticity in this size-response interaction.

With the present study, we aimed at replicating and extending our previous findings. Specifically, and based on the proposition that magnitude processing and motor control share a common representation (*Bueti & Walsh, 2009*; *Walsh, 2003*), we examined whether the compatibility effect between physical object size and space (i.e., horizontal response location) would also influence motor-related magnitude processing. To this end, we extended our paradigm to a visually-guided grasping task. Participants reached and grasped for a centrally positioned small or large object (a cube of 8 cm$^3$ or 64 cm$^3$) using either their left or right hand.

We made the following predictions: first, we expected to replicate the compatibility effect between physical object size and horizontal response location (*Ren et al., 2011*; *Wühr & Seegelke, 2018*). That is, reaction times should be shorter for small-left than for large-left associations and shorter for large-right than for small-right associations. Second, given that areas within the PPC are concerned with magnitude processing as well as sensorimotor transformations of spatial parameters required for specific motor actions (*Gallivan & Culham, 2015*; *Bueti & Walsh, 2009*), we expected that the compatibility effect should not only be present in reaction times, but also influence kinematic parameters of an action, particularly size-related parameters (i.e., latency and amplitude of maximal grip aperture). Third, in the context of numerical-spatial interactions, it has been reasoned that interaction effects between different magnitudes might be stronger in a visually guided motor task
compared to a keypress response task due to the stronger involvement of sensorimotor processing within parietal areas in such a task (*Fias, Lauwereyns & Lammertyn, 2001*). Consequently, we expected that the compatibility effect should be stronger in the present study compared to our previous study (*Wühr & Seegelke, 2018*).

## METHODS

### Participants

Based on previous research (*Wühr & Seegelke, 2018*), we defined a target sample size of 24 right-handed participants. We collected data from 31 individuals from Bielefeld University because we excluded data from seven participants (see below). All participants had normal or corrected to normal vision, were physically and neurologically healthy, and received course credit in exchange for their participation. We removed five participants from analysis as they exhibited more than 20% erroneous trials during either the compatible or incompatible conditions. Furthermore, we removed one participant due to technical problems with kinematic data recording and one left-handed participant. The final sample thus consisted of 24 participants (mean age = 22.83 years; SD = 3.37 years, range = 19–32 years; 9 female, 15 male; mean handedness score = 98.41, SD = 5.40, range = 80–100; *Dragovic, 2004*). The study was approved by the Bielefeld University Ethics committee (Ethical Application Ref: 2017-114), and all the participants provided written informed consent in accordance with the Declaration of Helsinki.

### Apparatus

The experimental setup was positioned on a custom-made shelf (200 cm $\times$ 60 cm) at a height of 80 cm. It consisted of two square blocks made of PVC ($10 \times 10 \times 3$ cm) with centrally embedded disks (7 cm in diameter) which served as start button for the left and right hand, respectively. The start buttons were located at the front edge of the shelf and spaced 40 cm apart. Another square PVC block ($18 \times 18 \times 3$ cm) with a centrally embedded disk (14.5 cm in diameter) was placed 30 cm and centrally behind the start buttons and served as the object base. The manipulated objects were two black cubes (small object: $2 \times 2 \times 2$ cm; large object: $4 \times 4 \times 4$ cm) and weighed 62 and 178 g, respectively. The experimental procedure was controlled via Presentation$^{\circledR}$ (Neurobehavioral Systems, Berkeley, CA, USA). Participants wore head phones and custom-made visual occlusion glasses which could be rapidly (<8 ms) made transparent or opaque during the experiment.

An optimal motion capture system consisting of 10 Bonita cameras with a spatial precision of approximately 1 mm (Vicon Motion Systems, Oxford, UK) was used to record kinematic data at 200 Hz sampling frequency. Five retro-reflective markers (10 mm in diameter) were attached to the distal end of the third metacarpal (MCP), the styloid process of the radius (WRT), the styloid process of the ulna (WRP), the thumb nail (TB), and the index finger nail (IDX) of each hand. The motion capture system was synchronized with Presentation$^{\circledR}$.

### Procedure

After filling out the informed consent and handedness inventory, the retro-reflective markers were placed on each hand. Participants sat centrally in front of the experimental

setup at a distance so that they could comfortably reach the object. At the start of each trial, the glasses turned opaque and the experimenter then placed one of the objects at the center of the object base. The experimenter carefully paid attention that the participant was unable to identify the size of the object from the sound of placing it at the object base. After a verbal signal from the experimenter, participants closed their index finger and thumb and depressed the start buttons with the side of their palms. This triggered the presentation of a low tone (250 Hz, duration 500 ms), informing participants that a trial was initiated. After 1,000 ms, the occlusion glasses turned transparent and participants grasped the object from the object base with either the left or the right hand (depending on condition), lifted the object, and placed it back at the object base. Participants were instructed to perform the task as quickly and accurately as possible.

There were four different experimental conditions, resulting from factorial combination of each level of the two factors Response hand (left, right) and S-R mapping (compatible, incompatible). For the compatible conditions, participants were required to grasp the small object with the left hand and the large object with the right hand. This mapping was reversed for the incompatible conditions. The factor S-R mapping was blocked, and the order of presentation was counterbalanced across participants. Within each block, participants performed 60 trials (30 with each hand), yielding a total of 120 trials. To familiarize participants with the current mapping, they performed 10 practice trials prior to the first block and 20 practice trials prior to the second block.

## Data processing

We first reconstructed the 3D coordinates of the retro-reflective markers and labeled them manually. We interpolated missing data points using a cubic spline (for gaps ≤10 frames) or the pattern fill algorithm (for gaps >10 frames) in Vicon Nexus 1.8.5, and low-pass filtered the data using a second-order Butterworth filter with a cutoff frequency of 10 Hz. We used custom written MatLab scripts (The MathWorks Version R2015a; Natick, MA, USA) for further kinematic post-processing.

We restricted our kinematic analyses to the reach-to-grasp phase, which we defined as the time period between movement initiation and object lift. We calculated the wrist joint center (WJC) of each hand as the midpoint between WRT and WRP. We defined reaction time (RT) as the time between when the glasses turned transparent and movement onset, which we calculated as the time at which the resultant velocity of WJC reached 50 mm/s. We defined movement time (MT) as the time between movement onset and object lift, as registered by micro switches in the object base. Finally, we calculated grip aperture of each hand as the Euclidean distance of TB and IDX in 3D space.

We excluded trials from analysis in which participants initiated their movements too early (0.3%), too slow (0.1%), used the wrong hand (6.2%), or in which the 3D coordinates of the markers could not be reconstructed (1.6%). In addition, we excluded trials in which RTs were <200 ms or >750 ms (1.5%) and trials in which MTs were >1,000 ms (<0.1%).

## Statistical approach

We performed statistical analyses of the data with a Bayesian model equivalent to a frequentist two-way repeated measures ANOVA using JASP (version 0.8.6; *JASP Team,*

*2018*; *Wagenmakers et al., 2018a*). Bayesian hypotheses testing involves a comparison of at least two different models (e.g., a model assuming a particular main effect and a null model that states the absence of such an effect) and these models need to be fully specified. The prior model odds indicate the relative plausibility of the competing models before seeing the data. The analysis estimates the posterior model odds (i.e., the relative probability of the models after observing the data) given the relative predictive probabilities of the models and the prior odds. The emphasis of Bayesian hypothesis testing is on the change in odds from prior to posterior brought about by the data, a quantity referred to as the Bayes factor. Hence, a Bayes factor hypothesis test indicates under which model the observed data are most likely (*Wagenmakers, 2007*; *Wagenmakers et al., 2018b*). *Lee & Wagenmakers (2013)* provide a descriptive and approximate classification scheme of the evidential strength associated with the Bayes factor values (1 = no evidence; 1–3 = anecdotal evidence; 3–10 = moderate evidence; 10–30 = strong evidence; 30–100 = very strong evidence; >100 = extreme evidence).

Our analyses considered all possible model comparisons, and hence captured all main effects and their interaction of our factorial design (i.e., S-R mapping (compatible, incompatible) and response hand (left, right)). Analyses were conducted separately on the following dependent variables: RT, MT, peak velocity, time to peak velocity (in ms), maximal grip aperture, time to maximal grip aperture (in ms).

## RESULTS

### RTs

As shown in Fig. 1, mean RTs were shorter with the compatible than with the incompatible mapping (mean difference left hand = 34 ms; mean difference right hand = 21 ms). The analysis confirmed that compared to the Null model, the Compatibility model received the most support from the data. The Bayes factor was 42,257 in support of the Compatibility model, indicating that the observed data were about 42,000 times more likely under the Compatibility model than under the Null model. Adding the main effect of response hand or the main effect of response hand and the interaction term decreased the degree of this support by a factor of about 5 for the two main effect model (42,257/8,947) and by a factor of about 8 for the interaction model (42,257/5,093), respectively.

### MTs and kinematics

The descriptive statistics of MTs and kinematic parameters are summarized in Table S1. For the right hand, MTs were shorter with the compatible than the incompatible mapping (mean difference = 57 ms), whereas the opposite was true for the left hand (mean difference = 14 ms). Confirming this impression, analysis revealed that compared to the Null model, the only model that received substantial support from the data was the model that included both main effects and the interaction term (Bayes factor = 19,583).

Mean peak velocity values exhibit a similar pattern of results. For right hand responses, peak velocity was larger with the compatible mapping than with the incompatible mapping (mean difference = 82 mm/s), whereas for left hand responses peak velocity was larger with the incompatible mapping than with the compatible mapping (mean difference

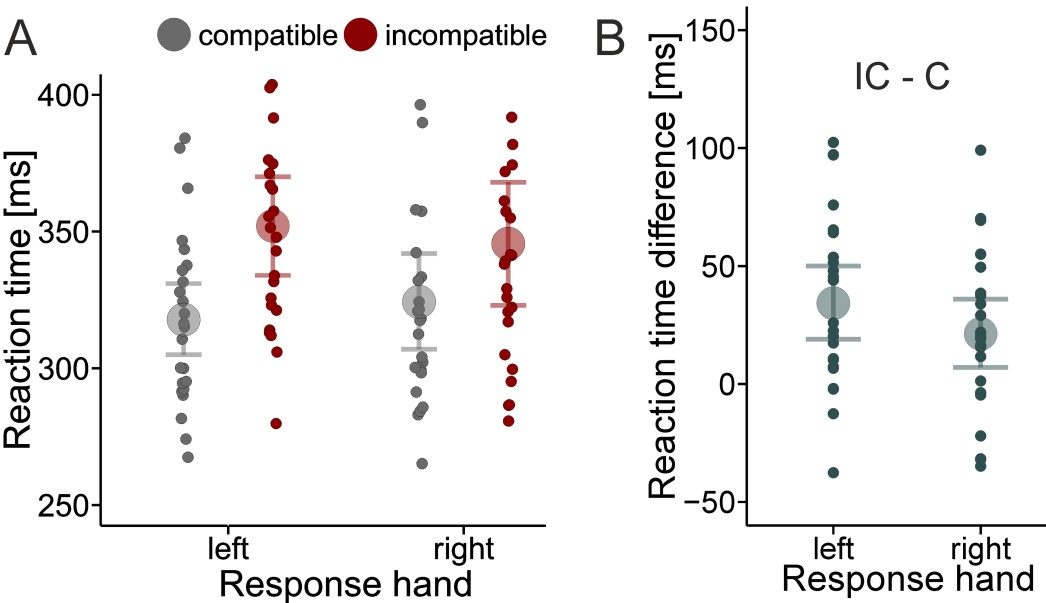

**Figure 1 Effect of compatibility on reaction times.** Group mean RTs (large dots) and individual mean RTs (small dots) as a function of S-R Compatibility and response hand (A). RT difference between the incompatible (IC) and compatible (C) mapping separately for each hand (B). Error bars reflect 95% credible intervals.

= 56 mm/s). As with MTs, compared to the Null model, the only model that received substantial support from the data was the full model (Bayes factor = 1977). Thus, these findings reflect that reach-to-grasp movements are often faster (i.e., shorter MTs and higher peak velocities) when grasping large as compared to small objects (*Seegelke et al., 2016*; *Castiello, Bennett & Stelmach, 1993*). Analysis on time to peak velocity showed that the data were best explained by the Null model or the Compatibility model. The Bayes factor was 1.42 (1/0.704) in favor of the Null model, and hence only indicating anecdotal evidence according to the classification scheme. All other models received considerably less evidence (Bayes factors between 5 and 15 in favor of the Null model).

Maximal grip aperture for right hand responses was larger with the compatible (mean = 102 mm) than the incompatible mapping (mean = 71 mm). In contrast, for left hand responses maximal grip aperture was larger with the incompatible (mean = 101 mm) than the compatible mapping (mean = 71 mm), indicating that maximal grip aperture scales with object size (cf. *Castiello, 2005*). The Bayes factor was 4.300e+48 in favor of the full model compared to the Null model. Furthermore, as shown in Fig. 2, maximal grip aperture was reached, on average, later with the incompatible mapping (388 ms) than with the compatible mapping (367 ms). This difference was more pronounced for the right hand (mean difference = 34 ms) than the left hand (mean difference = 8 ms). The analysis showed that both the Compatibility model and the full model received substantial evidence compared to the Null model (Bayes factor = 99.82 and 66.70, respectively). The evidence in favor of the Compatibility model over the full model was only a factor of 1.5
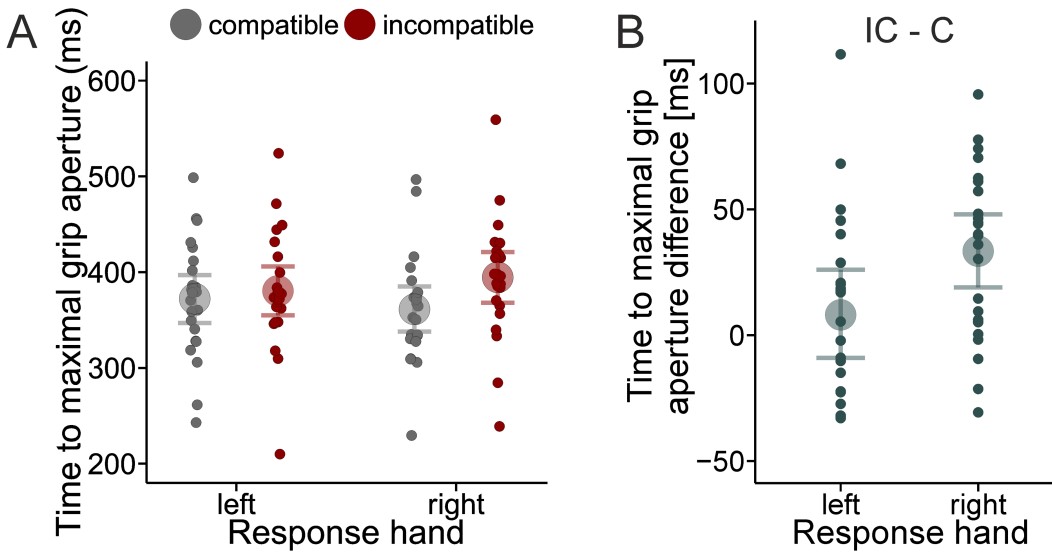

**Figure 2** **Effect of compatibility on the timing of grip aperture kinematics.** Group mean time to maximal grip aperture (large dots) and individual mean time to maximal grip aperture (small dots) as a function of S-R Compatibility and response hand (A). Time to maximal grip aperture difference between the incompatible (IC) and compatible (C) mapping separately for each hand (B). Error bars reflect 95% credible intervals.

(99.82/66.70). In sum the data demonstrate a strong compatibility effect that is evident in both RTs and kinematic parameters (i.e., time to maximal grip aperture).

## DISCUSSION

The present study examined size-space interactions during the performance of a visuomotor task. Participants reached and grasped a small or a large cube with either their non-dominant left or their dominant right hand. Our results revealed a compatibility effect between physical object size and horizontal response position (i.e., response hand). Specifically, participants initiated left hand movements faster when grasping the small cube compared to the large cube, whereas they initiated right hand movements faster when grasping the large cube compared to the small cube.

These findings thus replicate and extend findings from previous studies (*Ren et al., 2011*; *Wühr & Seegelke, 2018*) and demonstrate for the first time that the small-left/ large-right association between physical object size and response position is also present when the physical object size is clearly task-relevant. Furthermore, the compatibility effect also influenced the timing of grip aperture kinematics. On average, maximal grip aperture was reached earlier during compatible compared to incompatible trials. Together, these findings support the idea of a common neural metric underlying magnitude processing and sensorimotor control (*Bueti & Walsh, 2009*; *Walsh, 2003*; *Walsh, 2015*). An intriguing and open question which cannot easily be answered by ATOM is the direction of the observed compatibility effect. ATOM assumes some monotonic mapping of quantities, that is, more in one domain should correlate with more in another domain (*Walsh, 2015*; *Bueti & Walsh,*

*2009*). However, this concept cannot be readily applied to the (horizontal) spatial domain (i.e., left and right). Similarly, in the context of numerical-spatial interactions (e.g., SNARC effect), the origin of the orientation of the mental number line (MNL) remains a debated topic (*Rugani & Sartori, 2016*).

It has been suggested that the MNL originated from several cultural habits such as reading and writing direction (*Dehaene, Bossini & Giraux, 1993*; *Zebian, 2005*; *Shaki & Fischer, 2008*; *Shaki, Fischer & Göbel, 2012*; *Shaki, Fischer & Petrusic, 2009*) or finger counting direction (*Fischer, 2008*). However, numerical-spatial interactions have also been observed in preverbal infants (*Hevia & Spelke, 2010*; *Hevia et al., 2014*; *Bulf, Hevia & Macchi Cassia, 2016*), non-human primates (*Adachi, 2014*; *Drucker & Brannon, 2014*), and even birds (*Rugani et al., 2015*; *Rugani et al., 2011*). For example, even 3-day old domestic chicks, after being familiarized with a target number, associated a smaller number with left space and a larger number with right space to obtain a food reward (*Rugani et al., 2015*). Given that the numerical magnitude influenced the chicks' response selection (i.e., go to the left vs. go to the right), it might constitute a link between (numerical) magnitude processing and action (*Rugani & Sartori, 2016*). These findings suggest that the MNL, rather than being "culturally learned", originated from pre-linguistic and biologically determined precursors, maybe imposed by hemispheric asymmetries related to visuospatial attention (*Rugani et al., 2015*). Similarly, the compatibility effect between physical object size and response hand observed in the present and previous studies (*Ren et al., 2011*; *Wühr & Seegelke, 2018*) might be related to hemispheric lateralization as well, though the sources might be different. There are marked differences in the performance capabilities between the two hands (i.e., manual asymmetries (cf. *Goble & Brown, 2008*) ). For example, in right-handers, the dominant hand can produce greater forces than the non-dominant left hand (*Petersen et al., 1989*; *Armstrong & Oldham, 1999*; *Incel et al., 2002*), and this asymmetry is already present in childhood (*Hepping et al., 2015*). Consequently, this strength difference could entail a preference to grasp and lift larger (and heavier) objects with the dominant (right) hand than the non-dominant (left) hand.

We also found that, besides reaction times, the compatibility effect influenced the timing of grip aperture kinematics. Specifically, maximum grip aperture was reached later during incompatible trials compared to compatible trials for both the right and the left hand. Previous studies that have examined interactions between magnitude processing and sensorimotor control have found that these effects are often evident in movement initiation times (e.g., *Lindemann et al., 2007*; *Badets & Pesenti, 2011*; *Moretto & Di Pellegrino, 2008*) or during the initial stages of the movement trajectory (*Andres et al., 2008*; *Glover et al., 2004*; *Glover & Dixon, 2002*; *Namdar et al., 2014*). Along with the interpretations from these studies, we propose that the effect of magnitude processing on the timing of grip aperture in the present study reflects an interaction during motor planning stages, which typically occur before movement onset (*Wong, Haith & Krakauer, 2015*). However, motor planning is not considered a single and unified building block but is comprised of several sub-stages such as the choice and the description of the motion of the effector and the specification

of the motor command (*Wong, Haith & Krakauer, 2015*). Hence, it is still unclear at what exact stage the interaction originates—a question on which we will elaborate after the next paragraph.

Counter to our expectation, the compatibility effect did not influence maximal grip aperture. A potential reason for this null effect might be that participants adjusted their grip scaling during action execution. Although grip aperture is at least partly determined before movement onset (*Jakobson & Goodale, 1991*), maximal grip aperture is typically smaller when movements are executed in the presence of vision compared to when online vision is prevented (e.g., *Fukui & Inui, 2006*; *Seegelke et al., 2016*). Given that in the present study maximal grip aperture occurred clearly beyond the time necessary to utilize visual feedback (cf. *Elliott, Helsen & Chua, 2001*), it is possible that participants used visual feedback for online-corrections of their grip scaling.

We had further reasoned that the compatibility effect should be stronger in the present study compared to our previous study (*Wühr & Seegelke, 2018*). This prediction was based on the proposal that (numerical-spatial) interference effects (i.e., SNARC) should be more pronounced during tasks that recruit parietal areas (for example visuomotor tasks) and thus exhibit more neural overlap with magnitude processing (*Fias, Lauwereyns & Lammertyn, 2001*). In the study of *Fias, Lauwereyns & Lammertyn (2001)*, participants were required to respond to stimulus attributes that were more (orientation) or less (shape, color) associated with parietal cortex while ignoring concurrently presented digits. The authors found a SNARC effect (i.e., faster left responses in the presence of task-irrelevant small numbers, faster right responses in presence of task-irrelevant large numbers) for the orientation based judgments only.

In another study (*Badets et al., 2007*), participants judged the graspability of rods of different lengths. Participants overestimated their grasp when the presentation of the rod was preceded by a small number. Conversely, participants underestimated their grasp when a large number preceded the presentation of the rod. In contrast, when participants only compared the length of two successively presented rods (a pure perceptual judgment that does not involve any motor processes), numerical magnitude had no influence on the perceptual size judgment. Together, these results suggest that magnitude-interaction effects are present only (or at least more pronounced) when sensorimotor processes are involved, probably due to the larger extent of overlap in neural substrates in parietal areas.

However, counter to that reasoning, the compatibility effect in the present reaching and grasping task (mean RT difference of 27.5 ms between incompatible and compatible conditions) was of similar size compared to our previous button press task (30 ms on average; (*Wühr & Seegelke, 2018*). One possible explanation is that the association between physical size and horizontal location originates (mainly) at relatively early motor planning stages that are concerned with effector selection (i.e., using the left vs. the right hand) rather than later stages that are concerned with the specification of movement parameters (*Wong, Haith & Krakauer, 2015*). As the decision of what hand to choose was the same in the task of the present study as well as in our previous study (*Wühr & Seegelke, 2018*), it would seem rather surprising to observe any considerable differences. Of course, this interpretation is rather speculative and it remains certainly possible that task-related differences between

the two studies (e.g., with respect to stimuli, task conceptualization, participants, etc.) might have prevented the presence of a stronger effect in the present study.

## CONCLUSION

In summary, we demonstrated an S-R compatibility effect between stimulus size and response side (i.e., responding hand) in the RTs and in the kinematic parameters of movements for grasping real objects of different size. These findings suggest that the interaction between object size and response hand affects the planning of grasping movements and supports the notion of a strong overlap between the cognitive representation of (object) size and spatial (response) parameters, consistent with ATOM (*Walsh, 2003*; *Walsh, 2015*).

## ACKNOWLEDGEMENTS

We thank Agnieszka Keblowska for her help in data collection and data processing.

### Funding

We received support for the article processing charge from the Deutsche Forschungsgemeinschaft and the Open Access Publication Fund of Bielefeld University. The funders had no role in study design, data collection and analysis, decision to publish, or preparation of the manuscript.

### Grant Disclosures

The following grant information was disclosed by the authors:
Deutsche Forschungsgemeinschaft.
Bielefeld University.

### Competing Interests

The authors declare there are no competing interests.

### Author Contributions

- Christian Seegelke conceived and designed the experiments, performed the experiments, analyzed the data, contributed reagents/materials/analysis tools, prepared figures and/or tables, authored or reviewed drafts of the paper, approved the final draft.
- Peter Wühr conceived and designed the experiments, authored or reviewed drafts of the paper, approved the final draft.

### Human Ethics

The following information was supplied relating to ethical approvals (i.e., approving body and any reference numbers):

Bielefeld University granted Ethical approval to carry out the study within its facilities (Ethical Application Ref: 2017-114).

## Data Availability

Raw data, descriptive statistics, and outputs from the Bayesian analyses are provided as Supplemental Files.

## Supplemental Information

Supplemental information for this article can be found online at http://dx.doi.org/10.7717/peerj.6026#supplemental-information.

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
