# Peer review of "Compatibility between object size and response side in grasping: the left hand prefers smaller objects, the right hand prefers larger objects"

_PeerJ, doi:10.7717/peerj.6026_

## Round 0.1 · original submission · Minor Revisions

I have now received two reviews on your manuscript. I thank the reviewers for their work. As you will see, both reviewers are very positively inclined toward your work, and I share their view. I invite you to address the reviewers' comments. I especially invite you

a. to better specify the reasons underlying the choice of kinematics and the specific predictions related to each kinematics parameter (see comments of Reviewer 2),

b. to better ground your work in current literature, adding some further references (see suggestions of Reviewer 2)

c. to revise the figures and the English (see comments of Reviewer 1).

Thank you for sending your interesting work to PeerJ.

·

Basic reporting

Overall very clear and professionally written. Only a few minor comments (mostly English use) as follows:

73,90 - 'spatial aspects', 'motor aspects' - not clear what is meant. 'parameters'?
82 - 'trajectories shifts' - should be 'trajectory'
86 - 'movements'
113 - spell-out 'S-R'
121 - add ',' after 'specifically'
129 - 'movement latencies' - reaction times or overall movement times? Perhaps use 'lower' and 'higher' latencies to distinguish from object size?
135&136 - 'stronger' - compared to what?
146 - was this '20%' criterion set in advance? what is specific about 20% - is it, for example, the cut-off for chance performance?
149 - add 'years' after age stats
159 - add 'the' before 'object'
160 - change 'weighted' to 'weighed'
163 - 'rapidly' - how rapid? I am aware of similar technology (Plato goggles) which is very rapid (<ms)
165 - '1mm' - this must be at a specific distance from the camera
182 - insert 'as' before 'possible'
205-206 - change 'to' to 'too'
232 - remove 'considerably'
294 - please repeat 'mental number line (MNL)' here, as I'd forgotten what this was after reading results (retroacronym interference!)
317 - 'objects'; add 'than the non-dominant (left) hand'
339 - add 'the' before 'presence'
365 - remove [we demonstrated...] 'not only...artificial stimuli' - the authors did not present any new evidence showing this! Just make conclusions about the current data set, and refer to others' results for the symbolic stimuli

Figure 1 - Does left and right panel need A), B) labels? No need for colour really; Otherwise, very nice!
Figure 2 - (see Fig 1 commentS). Why not present the mean differences in Panel B as well - this the (second) critical effect?

Experimental design

No comment.

Validity of the findings

No comment.

Additional comments

Clear, concise, well-reported and very well analysed, really nice figures showing (most or) all of the necessary data.

Reviewer 2 ·

Basic reporting

Literature references provided is extend and well discussed. In my opinon, beside the studies of Lindemann et al. 2007 and Andres et al. 2008 (see lines 95-101), some references should be added: e.g. Badets and Pesenti (2010, Cognition) and Ranzini et al. (2011, Frontiers in Human Neuroscience).

Experimental design

The research question is clearly define and the knowledge gap being investigate, first of all in relation to the previous study of Wühr and Seegelke (2018), is identified and well discussed.
Since the present study, compared to Wühr and Seegelke (2018), added kinematic parameters, in my opinion it could be very relevant for the reader to understand the rationale behind these measures. In other words, I suggest authors to specify better the hypotheses expected as regard the specific kinematic parameters detected.
This could be very helpful also for understanding the findings and the respective discussion.

Validity of the findings

If I understand right the results' section, findings demonstrated a compatibilty effect only for RTs and for the time to maximal grip aperture.
I think that authors should explain why in the other measures (MTs, peak velocity, time peak velocity, maximal grip aperture) the overall compatibility effect did not emerged and, in some cases, the effect is reversed for the left hand.

Additional comments

In my opinion this study is a very interesting and meaningful replication of previous studies (Wühr & Seegelke, 2018; Ren et al. 2011) in which authors added two important aspects: a visually-guided grasping task and the measure of kinematic parameters.
While the first aspect is motivated (authors hypothesized that "interaction effects between different magnitudes might be stronger in a visually guided motor task...") and the finding is well discussed in the general discussion, the second aspect is less deepened (see my comments in the previous sections).

---

## Round 0.2 · accepted · Accept

I am happy to inform you that your paper has been accepted for publication on PeerJ.

#